# Common Features in Compulsive Sexual Behavior, Substance Use Disorders, Personality, Temperament, and Attachment—A Narrative Review

**DOI:** 10.3390/ijerph19010296

**Published:** 2021-12-28

**Authors:** Yaniv Efrati, Shane W. Kraus, Gal Kaplan

**Affiliations:** 1Faculty of Education, Bar-Ilan University, Ramat Gan 5290002, Israel; 2Psychology Department, University of Nevada, Las Vegas, NV 89154, USA; shane.kraus@unlv.edu; 3Baruch Ivcher School of Psychology, Reichman University, Herzliya 4610101, Israel; gal.kaplan@post.idc.ac.il

**Keywords:** compulsive sexual behavior, substance use disorder, big five personality, temperament, attachment orientations

## Abstract

Do addictions share common traits of an “addictive personality” or do different addictions have distinct personality profiles? This narrative review examines the differences in the associations between substance use disorder (SUD) and compulsive sexual behavior disorder (CSBD), on the one hand, and personality traits, attachment dispositions, and temperament, on the other hand. We found that both people with a SUD and people with CSBD tended to be more spontaneous, careless, and less reliable, to place self-interest above getting along with others, to show emotional instability and experience negative emotions such as anger, anxiety, and/or depression, to be less able to control their attention and/or behavior, and to be engulfed with a constant sensation of “wanting”. Only people with CSBD, but not SUD, noted concerns with their social ties, fear of losing close others, and/or trusting others around them. Results also suggested that people with a SUD and people with CSBD share high commonalities in personality traits and temperament, yet there are noted differences in their social tendencies, especially with close others. People with CSBD reported more concerns with possible relationship losses compared to people with SUD issues, who may be more worried about losing their source of escapism.

## 1. Introduction

Laypeople often relate the definition of addiction to substance use disorder (SUD). Research, however, has indicated that addictions do not only comprise the consumption of exogenous psychoactive substances such as drugs or alcohol, but also include behaviors. Non-substance behavioral addictions include internet addiction, internet gaming disorder, gambling disorder (formerly known as pathological gambling), compulsive buying, exercise dependence, food addiction, work addiction, and compulsive sexual behavior (often referred in the academic and popular culture as “sexual addiction”) [1,2]. Prior work has already examined whether there is an “addictive personality” that predisposes people to various forms of addictions [3,4], or whether there are different personality dispositions that predispose people to different forms of addictions. In the current review, we will draw similarities between SUD and one common behavioral addiction—compulsive sexual behavior disorder (CSBD)—in various trait-like dispositions—the big-five personality traits, attachment styles, and temperament. To date, most studies examined SUD and behavioral addiction separately [5,6] with only a few studies contrasting SUD and CSBD directly [4,7].

### SUD and Compulsive Sexual Behavior

SUD is a psychiatric disorder characterized by a pathological and compulsive pattern of substance-seeking and substance-taking behaviors that occupy most of a person’s time and efforts, leading to significant functional impairments at work, school, and/or home [8]. Persistent use of psychoactive drugs may lead to long-term changes in the brain (i.e., the development of new reward pathways), leading to multiple symptoms and features of addictions, including craving, withdrawal, and tolerance [9,10]. The disruptive pattern of substance-seeking behaviors persists despite the negative consequences of addiction (e.g., relationship, legal, health), with many individuals struggling to reduce or abstain from substance use [11,12]. Definitions of behavioral addictions, such as CSBD, share many commonalities with the definition of SUD, though this area still remains largely unexplored as new research continues to elucidate the etiology of this condition [13].

CSBD is defined as an impulse control disorder [14] characterized by a repetitive and intense preoccupation with sexual fantasies, urges, and behaviors, as well as an extensive pornography use and masturbation, use of paid sexual services, and risky sexual behaviors leading to clinically significant distress or impairment in social and occupational functioning and to other adverse consequences [14,15,16,17]. Although CSBD is officially classified as an impulse control disorder, the psychiatric classification of CSBD is still hotly debated [13,18]. In addition, most therapeutic interventions treat CSBD as an addiction [19], because it shares many addiction-like neurocognitive mechanisms and clinical characteristics [20]; however, despite some evidence of shared similarities, clinical studies examining the neurobiology of CSBD remain scarce.

Thus far, research on SUD and CSBD indicates that compulsive behavior, craving, and extensive preoccupation with the condition, which leads to impaired psychosocial functioning, are central characteristics in both addictions [21,22]. Do these disorders share commonalities in personality dispositions because they are all a result of an “addictive personality” that confers an inclination to addiction [23], or is the reason that one person is addicted to drugs (i.e., has SUD) and another to a sexual behavior a different personality profile of the affected person? To answer this question, we conducted a review on the correlation between SUD and CSBD, on the one hand, and three common personality-like dispositions—big five personality traits, attachment orientations, and temperament, on the other.

One of the most used personality classifications is known as the Five Factor Model (FFM) [24]. This classification emerged out of a series of attempts to understand the organization of trait descriptors in natural language [25,26,27]. Structural analyses of these descriptors consistently revealed five broad factors: extraversion (outgoing/energetic vs. solitary/reserved), agreeableness (friendly/compassionate vs. challenging/detached), conscientiousness (efficient/organized vs. easy-going/careless), neuroticism (sensitive/nervous vs. secure/confident), and openness to experience (inventive/curious vs. consistent/cautious). This structure has proven to be remarkably robust, with the same five factors observed in both self- and peer-ratings [28], in analyses of both children and adults [29], and across a wide variety of languages and cultures [30,31]. The big five personality traits have been extensively studied in relation with various addictions including SUD, though less literature is available on its relationship with CSBD.

Another domain in which people with SUD and people with CSBD might share commonalities is their temperament—individual differences in behavior that unlike personality are believed to be innate and relatively independent of learning, systems of values, and attitudes. One common classification of temperament perceives it as a 4-facet construct: negative affect including the inborn tendency for fear, sadness, discomfort, and frustration; effortful control comprising the innate ability for attentional, inhibitory and/or activation control; extraversion/surgency consisting of the inborn tendency for sociability, positive affect, and high-intensity pleasure; and orienting sensitivity comprising the innate neural and affective perceptual sensitivity. The dimension of “effortful control” received much attention in the etiology of psychopathology [32] and specifically of addictive behavior [33]. High effortful control includes the abilities to voluntarily manage attention (attentional regulation) and inhibit (inhibitory control) or activate (activational control) behavior as needed to adapt. For example, the abilities to focus attention when there are distractions, to not interrupt others and sit still in class or a movie theater, and to force oneself to do a tedious task are aspects of effortful control.

A second common classification of temperament relates to Cloninger’s biosocial model of personality [34] which emphasizes the biological background of personality. According to the model, personality could be assessed by four heritable, temperament dimensions of personality, and three additional character dimensions. The four temperament dimensions are: novelty seeking, comprising the innate tendency to respond actively to novel stimuli (a tendency supported by a correlation between novelty seeking and heighten functioning of the dopaminergic system); harm avoidance, consisting of the innate tendency towards an inhibitory response to signals of aversive stimuli leading to avoidance of punishment (often related to the functioning of the serotonergic system); reward dependence, comprising the inborn tendency for positive responses to signals of rewards in maintaining behaviors (often associated with the functioning of noradrenergic system); and, lastly, persistence, consisting of the inborn ability to maintain behaviors despite frustration and fatigue.

The three character dimensions are: self-directedness, comprising the ability of an individual to control, regulate and adapt one’s behavior in accordance with chosen goals and values; cooperativeness, consisting of the tendency towards social tolerance, empathy, helpfulness, and compassion; and self-transcendence, comprising the tendency for spirituality.

A final domain in which people with SUD and people with CSBD might share commonalities is people’s attachment orientations—trait-like dispositions that relate to social tendencies and emotion and stress regulation.

Attachment orientations are shaped during infancy via intimate interactions with caregivers in times of need [35]. When caregivers lend support and care, and the needs for comfort and security are consistently satisfied, the infant develops a secure bond towards the attachment figure (i.e., attachment security), which is characterized by a view of the self as lovable and of others as dependable. Secure people are generally more social and tend to develop healthy ties with family members, friends, and romantic partners.

At times, however, parental support is insufficient, and, as a result, infants might develop insecure attachment orientations that are classified along two dimensions, referred to as attachment anxiety and avoidance [36,37]. If infants’ needs are not sufficiently met by caregivers and the availability of support and care is uncertain, fear of abandonment can develop alongside an internalized anxiety of being rejected. Individuals with this attachment orientation are called anxiously attached and are characterized by an unfulfilled need for affection regardless of the amount of affection they receive [38]. If infants’ needs are not fulfilled and met with cold and distancing caregiving, infants will view others as untrustworthy and undependable and develop an attachment avoidance orientation. These individuals do not trust the goodwill of others and prefer to emotionally distance themselves from intimate relationships [39].

Research has indicated that attachment insecurities (both anxiety and avoidance) are associated with a general vulnerability to mental disorders [40,41]. For example, attachment insecurities are correlated with depression [42], generalized anxiety disorder [43], obsessive–compulsive disorder [44], posttraumatic stress disorder (PTSD) [45], eating disorders [46], and suicide ideation [47]. Therefore, attachment insecurities have transdiagnostic characteristics [48,49] and should be explored as they relate to behavioral disorders such as CSBD. Therefore, the goal of the current review is to examine the commonalities and differences in the big five personality traits, temperament, and attachment orientations between people who have a SUD and those with CSBD.

## 2. Methodology

### Search Strategy

The current review was conducted in accordance with the Preferred Reporting Items for Systematic Reviews and Meta-Analyses (PRISMA) guidelines. The process is presented in Figure 1. We conducted an electronic search for literature updated from 1 January 2000 to 30 November 2020. To identify relevant studies, we conducted two separate searches in the following six online databases: PsycINFO, PsychARTICLES, Open Grey, PubMed, Web of Science, and Psychiatric Abstracts (PubPsych, which includes PSYNDEX; PASCAL; ISOC-Psicologa; MEDLINE^®^; ERIC; NARCIS; NORART; PsychOpen; and PsychData). The first search was conducted on the association between CSB and personality-related aspects (i.e., attachment styles, the big five personality model, and temperament), and the second between SUD and personality-related aspects. In the first search, we used the following searching terms: (“Sex addiction” or “Hypersexual” or “Compulsive sexual behavior” or “Compulsive sexual behavior disorder” or “Compulsive sexual behaviour” or “Compulsive sexual behaviour disorder” or “CSBD” or “CSB”) AND (“big five Personality” or “five factor model” or “Temperament” or “Attachment Style” or “Attachment Orientation”). In the second search, we used the following searching terms: (“Drug addiction” or “Drug abuse” or “drug dependence” or “substance use disorder” or “substance dependence”) AND (“big five Personality” or “five factor model” or “Temperament” or “Attachment Style” or “Attachment Orientation”). Additionally, a snowball search was also conducted within Google Scholar to identify further studies which did not appear in the initial search.

After combining the two sets of database searches and the snowball search, and after removing duplicates, we were left with 739 articles. We then screened articles by abstract and title and screened by full-text evaluation according to the inclusion criteria. We included studies: (1) addressing the association between CSBD and personality-related aspects (i.e., attachment styles, the big five personality model, and temperament), or the association between SUD and personality-related aspects; (2) including 27 subjects or more for a clinical sample and 150 or more for a non-clinical sample; (3) based on a quantitative research design; and (4) published in the English language. Next, we excluded results that were derived from: (1) conferences or not a published article or chapters; (2) individuals younger than 16 years of age; (3) sex offenders; (4) individuals with psychiatric disorders other than CSBD and SUD; (5) animal models (i.e., non-human participants); (6) addictions that are not substance related; and/or (7) participants with comorbid psychiatric disorders. This screening process led to 88 potentially eligible papers. After reviewing the articles’ full text, 19 additional papers were excluded (see Figure 1) for a total of 69 papers in the final review. Methodology reported based on Orilisi et al. [50].

## 3. Results

### 3.1. Commonalities and Differences in Big Five Personality Dispositions

To date, a few dozen studies have been published on the correlation between SUD and the big five personality traits [6,51,52]. In contrast, only 9 studies correlated compulsive sexual behavior and the big five personality traits [1,2,5,53,54,55,56,57], and only one study compared SUD and people with compulsive sexual behavior directly [4]. Table 1 summarizes the main findings.

Zilberman and colleagues [4] revealed that the “addictive personality” of people with compulsive sexual behavior and those with SUD is remarkedly similar with respect to the five major facets of personality. Both groups score low on agreeableness and conscientiousness and high on neuroticism. Research that has separately examined the big five traits among people with a SUD and people with compulsive sexual behavior shows high agreement with Zilberman and colleagues’ [4] findings. Specifically, in 8 out of 9 studies on compulsive sexual behavior and personality dispositions, compulsive sexual behavior was correlated with higher neuroticism; 7 out of 9 studies correlated it with lower conscientiousness, and 4 out of 9 with lower agreeableness. These seemingly robust results share high commonality with studies on the correlation between SUD and personality dispositions such that all studies show that SUD is correlated with higher neuroticism and lower conscientiousness; most studies (but not all) related SUD with lower agreeableness.

This pattern of results also indicates that people with a SUD and people with compulsive sexual behavior tend to be more spontaneous, careless, and less reliable (i.e., low conscientiousness), to place self-interest above getting along with others (i.e., low agreeableness), and to show greater emotional instability and experience negative emotions, such as anger, anxiety, and/or depression (i.e., high neuroticism).

The final two personality dispositions—openness to experience and extraversion—were only sporadically correlated with addictive behavior. Whereas openness to experience was not reliably correlated with addictive behavior, there are inconsistencies regarding extraversion in both groups (i.e., compulsive sexual behavior and SUD). A meta-analysis covering 175 studies (published until 2007) on SUD and the big five personality traits indicated that people with a SUD are lower on extraversion—i.e., are more solitary and reserved. However, studies that were published since then did not reveal significant associations between SUD and extraversion. Similarly, the findings on people with compulsive sexual behavior are also equivocal such that some do not find any association [4,5,53,55,56,63], one found lower extraversion [61], and two, higher [54,57]. Thus, a meta-analysis regarding the correlation with extraversion is warranted to examine the role of extraversion in addictions. Here, we would like to suggest one possible moderator that might explain the inconsistent correlation between CSBD and extraversion—the type of CSBD in question.

### 3.2. Commonalities and Differences in Temperament

Research on the association between the 4 facets of temperament (negative affect, effortful control, extraversion/surgency, and orienting sensitivity) and addictive behavior highlighted effortful control as a key player in addictions. The current reviewed studies agree with this finding (see Table 2 and Table 3). Specifically, studies examining the role of effortful control in SUD found that low effortful control (i.e., lower ability to regulate or control behaviors) reliably relates to SUD at all stages of addiction [33,74,75,76,77,78,79]. High effortful control was correlated with less SUD [80] and a lower drinking frequency [81]. For example, Santens and colleagues [82] conducted a study on 712 SUD adult patients and found that high effortful control was characteristic of the “resilient” group, whereas low effortful control was typical with the “anxious” and “reward-sensitive” groups. This review did not reveal any other associations between the remaining temperament clusters (e.g., negative affect, extraversion/surgency, and orienting sensitivity) and SUD.

To date, only a single study on 310 adolescents has examined the associations between temperament (i.e., negative affect, effortful control, extraversion/surgency, and orienting sensitivity) and CSBD [107] using the current classification of temperament. In accordance with the correlation between SUD and low effortful control, lower effortful control was found to be related to more severe symptoms of CSBD [107]. Furthermore, this study also noted positive associations between higher orienting sensitivity and CSBD. Because only a single study examined the association between temperament and CSBD, it is premature to draw conclusions regarding the role of orienting sensitivity.

Research on the association between Cloninger’s biosocial clusters of personality and SUD revealed one reliable heritable, temperament cluster—novelty seeking. People with an SUD seem to have an overly active dopaminergic system and thus a greater innate tendency to actively explore novel stimulation, while exhibiting impulsive decision making. The dopaminergic system is part of the reward system and relates to the sensation of “wanting” [108], which correlated perfectly with SUD. The single study that examined the association between CSBD and Cloninger’s biosocial clusters of personality [104] revealed a similar positive association with novelty seeking, which also correlated the compulsive engagement of people with CSBD with sexuality and the “wanting” sensation of sexual-related behaviors and/or cognitions [109].

The review also revealed two additional (somewhat) consistent associations between SUD: self-directedness (53% consistency) [84,86,87,88,90,91,94,98,102] and cooperativeness (53% consistency) [84,86,87,89,90,91,94,96,102]. People with an SUD are less able to control and/or regulate their behavior following chosen goals and values, or to demonstrate social tolerance, empathy, helpfulness, and compassion towards others. CSBD was also found to be related with lower self-directedness [104], which fits well with the inability to control one’s behavior (i.e., impaired control) and maintain a goal-directed path in one’s life. This tendency is also consistent with the association of SUD, CSBD, and low effortful control (i.e., less ability to control one’s attention and goal directed behavior).

### 3.3. Commonalities and Differences in Attachment Orientations

The associations between attachment orientations and SUD have been extensively examined in cross-sectional and longitudinal studies [110]. In a meta-analysis summarizing 34 longitudinal samples and comprising 56,721 participants, a significant correlation was noted between attachment insecurity (both anxiety and avoidance) and SUD such that the higher people’s attachment anxiety and/or avoidance, the higher the likelihood for SUD. In addition, researchers found that, whereas earlier attachment insecurity predicts later SUD, earlier SUD does not predict later attachment insecurity—a finding supporting attachment insecurity as a predisposition for SUD and not an outcome of it. Of note, however, the correlations between insecure attachment orientations and SUD are generally weak to negligible, with a shared variance of approximately 2%. Recently, Estévez and colleagues [111] directly compared the associations between attachment orientations, SUD, and several behavioral addictions (problematic internet use, video game addictions, and gambling disorders; but did not include compulsive sexual behavior) and found that, whereas attachment insecurity reliably correlated with higher likelihood for all behavioral addictions, it was not, however, correlated with SUD.

In contrast to SUD, the associations between attachment orientations and CSBD are more indicated [5,105,112,113,114,115,116,117,118,119]. Specifically, attachment insecurity (both anxiety and avoidance) relates to greater likelihood of CSBD and higher symptom severity of CSBD, with a shared variance ranging from 5% to 21%. Thus, it seems possible that attachment insecurity that relates to various social dysfunctions, greater distress, and emotion dysregulation is a predisposition for addictions, but this could be particularly salient for development of CSBD.

## 4. Discussion

In sum, people with an SUD and people with compulsive sexual behavior tend to be more spontaneous, careless, and less reliable (i.e., low conscientiousness), to place self-interest above getting along with others (i.e., low agreeableness), to show emotional instability and experience negative emotions such as anger, anxiety, and/or depression (i.e., high neuroticism), to be less able to control their attention and/or behavior (i.e., low effortful control and self- directedness), and to be engulfed with a constant sensation of “wanting” (i.e., high novelty seeking). These correlational clusters may shed further light on our understanding of the psychology of addictive behaviors, particularly as it relates to possible differences between clinical and non-clinical populations. However, only people with compulsive sexual behavior, but not SUD, are especially concerned with their social ties, fear of losing close others, and/or trusting others around them. These latter differences seem to fit well with the differences in addiction type—addiction to a substance versus addiction to a behavior. Acknowledging these commonalities and differences may allow a better detection of risk factors attributed to the development of addictive behaviors and possibly offer better-suited therapies for people reporting issues with addictions. Further research is particularly needed to examine personality classification among individuals seeking treatment for CSBD given the considerable absence of clinical data. Moreover, additional research is needed to examine possible gender differences in personality classification, particularly as it relates to the clinical manifestation of CSBD, which remains still understudied. Moreover, examining personality classification within subtypes of CSBD (e.g., exclusive problematic pornography use, engagement in anonymous/casual sex with strangers or paid sex workers) could also elucidate possible personality differences between solitary and dyadic sexual behavior, which in turn could inform treatment strategies for help seeking individuals. In a similar vein, further research is also needed to examine the specific relationships between CSBD, personality dispositions, attachment style, and temperament as a function of substance use (e.g., stimulant, sedative, alcohol, cannabis) since such information could possibly help to identify possible symptoms and personality clusters in clinical and non-clinical populations.

## 5. Conclusions

In conclusion, we view this research as important in studying the personality classification (personality traits, temperament, and attachment dispositions) related to people with SUD or CSBD. These findings add to the body of data that may help to better understand the personality underpinnings of people with SUD or CSBD even when their symptoms are below the clinical threshold. In addition, the current research may help to better tailor interventions aimed at reducing SUD or CSBD and its negative outcomes by targeting specific personality classification considered highly indicative of SUD or CSBD.

## Figures and Tables

**Figure 1 ijerph-19-00296-f001:**
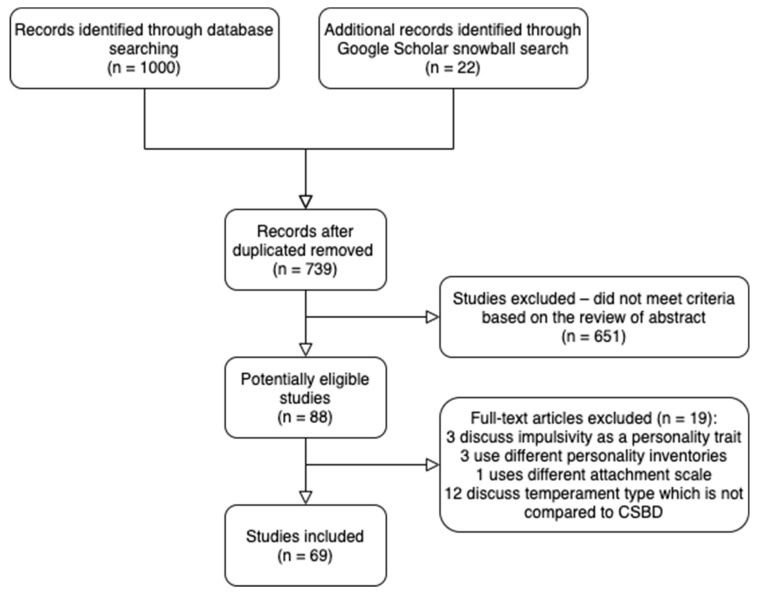
PRISMA flow chart regarding paper selection process (*n*= 1022).

**Table 1 ijerph-19-00296-t001:** Personality traits on CSBD and SUD.

Study	Design	Sample (*n*, Sex)	Mean Age (Year)	Measure	CSBD					Substance Abuse				
					A	C	O	N	E	A	C	O	N	E
Zilberman et al., 2018 [4]	CSS	Drugs (*n* = 58): 45 male and 13 Female. CSBD (*n* = 65): 57 male and 4 Female.	Age (mean rank): Drug- 149 CSBD- 132	Drug- DAST [58]. CSBD: Individual-based compulsive sexual behavior scale [59]	- 29.44 (5.74)	- 28.62 (6.078)	+ 36.89 (6.54)			- 28.67 (7)	- 29.88 (6.725)		+ 26.97 (5.68)	
Efrati & Gola [5]	CSS	618 Israeli adolescents (341 boys and 277 girls)	Aged 14–18 years (M = 16.69, SD = 1.16),	Individual-based compulsive sexual behavior scale [59]	- 3.37 [0.54]			+ 3.13 [0.62]						
Pinto et al., 2013 [53]	CSS	152 male college students recruited in a Portuguese university	22 years (standard deviation [SD] = 2.63) raging from 18 to 33	Compulsive Sexual Behavior Inventory; CSBI-22 [60]	- B = −0.35			+ B = 0.21						
Rettenberger et al., 2016 [61]	CSS	1749 German students, 56.5% (*n* = 988) were female, 42.9% (*n* = 750) male, and 0.6% (*n* = 11) described themselves as neither male nor female (e.g., transgender).	M = 24.42 (SD = 4.37, range 18–62).	Hypersexual Behavior Inventory; HBI [62]	- r = −0.05	+ r = −0.22		- r = 0.07						
Walton et al., 2017 [54]	CSS	510 Australian participants (267 males and 243 females)	The respective mean age of male and female participants was 36.52 years (SD = 12.66) and 30.38 years (SD = 12.12).	Hypersexual Behavior Inventory; HBI [62]	- b = −0.44	- b = −0.16		+ b = 0.44	+ b = 0.25					
Shimoni et al., 2018 [63]	CSS	267 Israeli participants (186 males and 81 females)	mean age of 30.2 years (*SD* = 9.8) and age range of 18–68.	Sexual Addiction Screening Test; SAST [64]		- β = −0.21	+ β = 0.18	+ β = 0.15						
Amamou et al., 2020 [55]	CSS	510 Tunisian volunteers. 360 women (60%) and 204 men (40%)	The average age was 31.5 +/− 9.3 years	Sexual Addiction Screening Test; SAST [64]		- 3.30 [0.64]		+ 3.61 [0.94]						
Paz et al., 2021 [56]	CSS	Israeli participants. The sample comprised solely of men. Of these, 113 identified themselves as heterosexuals (63.8%), 48 as gay men (27.1%), and the remaining 16 as bisexuals (9%).	The participants’ mean age was 32.44 years (SD = 8.41), ranging from 19 to 70 years.	Bergen–Yale Sex Addiction Scale; BYSAS [65]		- r = −0.152		+ r = 0.173						
Soraci et al., 2021 [57]	CSS	1230 Italian participants. (26.7% males, 73.1% females, other 0.2%)	Mean age 24.9 years [SD ± 5.60];	Bergen–Yale Sex Addiction Scale; BYSAS [65]			+ 7.01 [1.83]		+ 5.72 [2.43]					
Fehrman et al., 2019 [6]	CSS	1885 participants (male/female = 943/942)	18–24 years (643; 34.1%), 25–34 years (481; 25.5%), 35–44 years (356; 18.9%), 45–54 years (294; 15.6%), 55–64 (93; 4.9%), and over 65 (18; 1%).	Participants were questioned concerning their use of 18 legal and illegal drugs.						- ≤44–49	- ≤44–49	+ ≥51–56	+ ≥51–56	
Dash et al., 2019 [52]	CSS	Participants were 3785 twins and siblings from Australian Twin Registry (1365 men, 2420 women).	Age: M = 32 years, range 21–46 years	Australian version of the Semi-Structured Assessment of the Genetics of Alcoholism [66,67]						- 3.6 SE-0.01	- 3.7 SE-0.02		+ 2.47 SE-0.02	
Kotov et al., 2010 [51] based on studies up to 2007	CSS +LS	The review included 175 studies published from 1980 to 2007, which yielded 851 effect sizes. For a given analysis, the number of studies ranged from three to 63 (total sample size ranged from 1076 to 75,229).	N/A	Diagnoses were made by a trained rater according to one of the modern classification systems, namely the DSM–III, DSM–III–R, DSM–IV, ICD–9, ICD–10, or Research Diagnostic Criteria [68]						- d = −0.75	- d = −1.02		+ d = 1.13	- d = 0.33
Sattler & Schunck, 2016 [69]	CSS	German employees: *n* = 6454 (Male: 0.53)	Age: M = 40.63, SD = 8.64.	Cognitive Enhancement for drug use							- OR = 0.774		+ OR = 1.352	
Terracciano et al., 2008 [70]	CSS	1102 Participants from the East Baltimore (About 62% of the sample was female; 63%)	age ranged from 30 to 94 years (M = 56.6; SD = 12.4)	Classified: “never use”; “former use” (as those who use but not in the last seven days), and ‘current use’ as those who use in the last seven days							- 40.3 (1.77)		+ 57.6 (1.77)	
Lackner et al., 2013 [71]	CSS	63 Austrian male substance dependents (30 alcohol abusers, 33 polydrug abusers)	The alcohol abusers mean age was 42 years (SD = 8.54), whereas the polydrug abusers mean age was 31 years (SD = 8.39	Expert assessment (not further specified)						- d = −0.42	- d = −0.64	- d = −1.10	+ d = 0.64	
Raketic et al., 2017 [72]	CSS	62 woman outpatients from Serbia +30 control group. 30 women who had alcohol use disorder and 32 women who had opioid use disorder.	Opiate dependent (M = 35.4, SD = ±5.2); Alcohol dependent (M = 39.9, SD = ±5.1); Control group (M = 36.1, SD = ±5.6).	Expert assessment (not further specified)							- 165.0 [±16.4]		+ 148.0 [±21.6]	
Hwang et al., 2014 [73]	CSS	30 patients from Korea, diagnosed with alcohol dependence (mean age, 30.03 ± 5.89 years), and 30 healthy controls	Alcohol Dependence: (M = 30.03, SD = ±5.89 years); healthy controls (M = 25.33, SD = ±2.77 years).	Expert assessment (not further specified)						- 37.33 [6.09]	- 36.43 [9.93]		+ 39.07 [8.37]	- 36.37 [8.50]

*Note*. A = Agreeableness, C = Conscientiousness, O = Openness, *n* = Neuroticism, E = Extraversion, + = positive correlation, - = negative correlation; CSS = cross-sectional study; LS = longitudinal study; OR = Odds Ratios.

**Table 2 ijerph-19-00296-t002:** Temperament traits (TCI; Temperament and Character Inventory) on CSBD and substance abuse.

					Substance Abuse						
Study	Design	Sample (*n*, Sex)	Mean Age (Year)	Measure	NS	HA	RD	PS	SD	CO	ST
Bozkurt et al., 2014 [83]; Alcohol	CSS	*n* = 94 male alcohol-dependent inpatients and A healthy control group (*n =* 63).	male alcohol-dependent (M = 44.04), A healthy control group (M = 35.24)	The substance dependence section of the SCID-I [84]	+	+	-	-	-	-	
					r = 0.46	r = 0.45	r = −0.29	r = −0.36	r = −0.49	r = −0.48	
Tomassini et al., 2012 [85]; Alcohol	CSS	Twenty-seven abstinent alcohol-dependent subjects (21 males and 6 females	Age (M = 46.15, SD = ±7.67)	Expert assessment (not further specified)	-						
					r = −0.34						
Abassi et al., 2015 [86]; Morphine	CSS	120 Iranians with morphine (opioid) use disorder	Age of addicts was M = 36.45, SD = 4.37 year, with a range of 20–40 years old.	Expert assessment (not further specified)	+	+	-	-	-	-	
					r = 0.54	r = 0.33	r = −0.44	r = −0.41	r = −0.57	r = −0.52	
Can et al., 2014 [87]; Substance abuse	CSS	87 male substance abusers from Turkish and 50 healthy male volunteers	Age (M = 21.3, SD = ±2.3)	Expert assessment (not further specified)	+	+		-	-	-	
					21.5	1.6		3.7	18.9	17.9	
					[4.2]	[5.7]		[1.9]	[5.5]	[7.9]	
Hashemi et al., 2019 [88]; Drug	CSS	58 men and 52 women from addiction treatment clinics-Iran.	Men: (M = 36.00, SD = 7.66)	Expert assessment (not further specified)	+	+	-	-	-		
		Control group,	Woman: (M = 30.94)		8.74	8.04	5.61	2.5	8.79		
		58 men and 52 women	SD = 6.94.		[2.12]	[1.97]	[1.72]	[0.94]	[3.25]		
		from the general population									
Amirabadi et al., 2015 [89]; Opiate	CSS	45 male nicotine use disorder and 45 male opioid use disorder individuals	Opiate addicts M = 35.97, SD = 7.24) Nicotine addicts M = 39.02, SD = 6.22	Expert assessment (not further specified)	+					-	
					86.08 (8.51)					87.46 (6.52)	
Fassino et al., 2004 [90]; Heroin	CSS	180 heroin abusers. (83.3% of these were men, 16.7% were women)	Man: 31.38 years (SD = 6.06).	The Structured Clinical Interview; SCID II [83]	+	+	-		-	-	+
			Woman: 28.78 years (SD = 6.30).		20.92 (4.73)	14.98 (6.31)	15.05 (4.21)		30.29 (4.03)	29.07 (6.18)	14.56 (6.40)
Hosák et al., 2004 [91]; Methamphetamine	CSS	41 inpatients dependent on methamphetamine, and 35 controls.	Age (M = 24.0, SD = ±3.9) years. Range: 19–32. Women (n = 12).	Expert assessment (not further specified)	+	+		-	-	-	+
					26.8 [4.7]	18.0 [6.5]		3.9 [1.8]	20.2	26.0 [7.1]	18.2
									[6.2]		[6.7]
Steingrimsson et al., 2020 [92]; Drug	CSS	6917 individuals from Sweden (58% women)	N/A	The alcohol use disorders identification test; AUDIT [93].					-	-	
				Drug Use Disorders Identification Test; DUDIT [94]					r = −0.11	r = −0.09	
Chang et al., 2007 [95]; Substance abuse	CSS	60 males with history of substance abuse	Age: M = 17.68, SD = 1.45 years (range, 14−20 years)	Substance abuse index (SAI)	+						
					19.0 [3.4]						
Sarra et al., 2014 [96]; Substance abuse	CSS	84 participants from drug treatment services in Italy, 74 (88.1%) are males and 10 (11.9%) females.	M = 30	Expert assessment (not further specified)	+		+			+	-
			(minimum = 17, maximum = 47, SD = 7.5) years.		E = 0.319 [0.141]		E = 0.410 [0.168]			E = 0.853 [0.153]	E = −0.330 [0.165]
Lukasiewicz et al., 2008 [97]; Substance abuse	CSS	Alcohol abuse and dependence: *n* = 167 (95.2% Male), Drug abuse and dependence: *n*= 270 (93.0% Male).	Age: M = 39, SD = ± 13	Expert assessment (not further specified)	+						
		General population-N = 998 (90.1% Male).	Years.		11.2						
					[2.6]						
Watanabe et al., 2011 [98]; Substance abuse	CSS	3802 Japanese university students. 1109 men and 2693 women.	Age: M = 20.2, SD = 1.5.	Prevalence of smoking and alcohol use	+	-	+	-	-		
					r = 0.18	r = −0.13	r = 0.06	r = −0.09	r = −0.10		
Milivojevic et al., 2012 [99]; Opiate and Alcohol	CSS	Opiate addicts: 312 subjects from Serbia, 66 females and 246 males. Alcoholics: 100 subjects, 36 females and 64 males. Control group: 346 volunteers (177 females and 169 males).	Alcohol addicts: 39.21 years (SD 11.1), Opiate addicts 26.32 years (SD 5.99), and Normal controls 23.33 years (SD 6.79).	Clinical interview by a psychiatrist and DSM IV TR criteria.	+		-		-		+
					OR = 3.61 (2.68–4.86)		OR0 = 0.76 (0.60–0.96)		OR= 0.64 (0.46–0.88)		OR = 1.37 (1.08–1.71)
Conway et al., 2003 [100]; Substance abuse	CSS	326 addiction treatment from USA was 44.2% male, 54.9%	Age: 32.9 (SD = 7.9)	Lifetime history of most serious substance of dependence.	+						
					23.67−24.44						
Herrero et al., 2008 [101]; Cocaine	CSS	120 individuals from Spain, 38.3% were women.	Age: 23.8 years (SD = 3.4; range 18–31)	Preferred Reporting Items for Systematic Reviews and Meta-Analyses [102]					-	-	+
									41.9 (10.4)	42.3 (9.4)	53.8 (10.5)
			**Compulsive sexual behavior disorder**								
					ST	CO	SD	PS	RD	HA	NS
do Amaral et al., 2015 [103]	CSS	69 sexually compulsive MSM from Brazil	Age: M = 35.2, SD = 8.2.	Sexual Compulsivity Scale; SCS [104]	+				-		
					25.92 (5.22)				15.72 (6.36)		

*Note*. NS = Novelty Seeking, HA = Harm Avoidance, RD = Reward Dependence, PS = Persistence, SD = Self-Directedness, CO = Cooperativeness, ST = Self-Transcendence, + = positive correlation, - = negative correlation, CSS = cross-sectional study, E = Estimate.

**Table 3 ijerph-19-00296-t003:** Temperament traits (ATQ; Adult Temperament Questionnaire) on substance abuse and CSBD.

						Substance Abuse		
Studies	Design	Sample (*n*, Sex)	Mean Age (Year)	Measure	NA	EC	E/S	OS
Mun et al., 2018 [33]; Tobacco, alcohol, and marijuana	LS	311 adolescents with parents from USA.	LS-from 16 to 27	Composite International Diagnostic Interview [105]		-		
						r = −0.05		
Cheetham et al., 2010 [74]; Substance use	Review	N/A	N/A	N/A	+	-		
					N/A	N/A		
Nigg et al., 2004 [80]; Alcohol	LS	boys from 198 families	LS- followed between 3 and 14 years	Alcoholism subtype groups		-		
						N/A		
Wong et al., 2013 [81]; Alcohol and drug	CSS	644 undergraduate students (67.2% female)	23.58 (SD = 6.861)	Drinking and other Drug Use History Questionnaire; DDHQ [106]		-		+
						R = −0.86 OR = 0.42		r = 0.03 OR = 1.03
Santens et al., 2018 [82]; Substance use	CSS	700 adult Caucasian patients on treatment program for SUDs. (68.1% males and 31.9% females).	45.7 years (SD = 11.25).	Addicted group		-		
						M ≥ −0.500000		
						CSBD		
					NA	EC	E/S	OS
Efrati, 2018 [107]	CSS	310 high-school students (183 male, 127 female)	age 16 to 18 years (M = 16.94, SD = 0.65),	Individual-based compulsive sexual behavior scale [59]		-		+
						r = −0.11		r = 0.11

*Note*. NA = Negative affect, EC = Effortful control, E/S = Extraversion/Surgency, OS = Orienting sensitivity, + = positive correlation, − = negative correlation, CSS = Cross-sectional study; LS = Longitudinal study.

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
