# Peer review of "Common Features in Compulsive Sexual Behavior, Substance Use Disorders, Personality, Temperament, and Attachment—A Narrative Review"

_ijerph, 2021, doi:10.3390/ijerph19010296_

Round 1

Reviewer 1 Report

Broad Comments

a.       The ACS referencing style is supported by the journal, but several spelling of journal names should be corrected (e.g. “Addictive behavior” ->  “Addictive Behavior”). DOI-digits would be embraced.

b.       “Addictive personality” is not explained. Further, nowhere in the text any “personality profile” is given. I am afraid this concept is only used in its colloquial sense. The heading suggests that the appropriateness of the concept would be examined, instead correlational coherencies are reported – without referencing any authority.

c.       The article is not age- and not gender-sensitive.

d.       “Substance abuse” is not adequately used. Authors obviously understand this as an overarching term, but this would be the term “addiction” or rather “substance use disorder (SUD”). In DSM-5  it is “mild” or “moderate” or “severe” SUD, in DSM-IV it was “substance abuse” and “substance dependence”. In ICD-10 it was “harmful use” (pretty much the same as “abuse”) and “dependence syndrome” were sub-categories. So “substance abuse/ harmful use” would logically exclude “dependence”.

The literature search terms (line 183f.), again, indicate that search results are not affected by this; still the term “abuse” as used here might perplex readers (especially psychiatrists).

e.       Review design and presentations do not match the standards of an international journal. Authors’ review is not a ‘systematic review’ in terms of the PRISMA statement and other articles in the journal (e.g. Giulia Orilisi, Marco Mascitti, Lucrezia Togni, et al. Oral Manifestations of COVID-19 in Hospitalized Patients. Int. J. Environ. Res. Public Health 2021, 18(23), 12511; doi.org/10.3390/ijerph182312511.)

For a systematic review, too many information is missing – especially inclusion criteria, study quality assessment (best with results table), and table of descriptive characteristics of included studies. (See details below.)

f.        Authors’ conclusions are not adequately supported by their data. This affects interpretation (internal validity): It should be evidenced that the found correlations are specific for patients with compulsive sexual behavior (CSBD) and patients with substance use disorders (SUD) in contrast with non-clinical populations. And does the substance consumed make no difference? The specific substance/ substance group may be a moderator (sedatives, stimulants, multiple substance use).

g.       Generalizability (external validity) of their results: “Do all addictions share common traits of an ‘addictive personality’ or do different addictions have distinct personality profiles?” – So starts the Abstract, but that is not what the text is about. Logical failure: Authors only study substance-related addiction and leave out all behavior-related ones. Thus, they can never refer to “all” addictions. The proposition in the Discussion section (line 355f.) is therefore over-interpreted: “…these personality-related dispositions seem to be at a core of a ‘addictive personality’.”

h.       After all of this, the heading is wrong: “Common features in compulsive sexual behavior, substance use disorders, personality, temperament and attachment – a narrative review” would have been more appropriate.

Specific Comments

Spelling of the heading is incorrect (uses no capitals).

Affiliations of authors are only vaguely given: Which faculty or department? What is this Interdisciplinary Center? E-Mail address of authors usually is their institutional address.

After table notes should be an empty line.

Parts 245-262, 283-302, and § ‘3.3 Commonalities and differences in attachment orientations’ do not report results. They belong to the Discussion section.

Parts ‘4.1 Future Directions’ and ‘4.2 Therapeutic Implications of the Current Research’ are interesting per se but are  beyond the scope of the article.

Authors should limit the included studies to controlled studies to enhance internal validity of their findings (see Broad Comments f). Then they will have not 70 (but rather 17) studies to include.

The magnitudes of correlations are of SOME interest and should be reported in an international academic journal. 

Line

Text / remarks / suggestions / proposals

39

“forms addictions” -> forms of addictions

67

“classify  as” -> classified  as

74f.

Reference?

86

“natural language” -> spoken language?

100-130

Names of constructs/ traits/ attributes are sometimes written in capitals, sometimes not. In the rest of the text no capital letters were used for this.

148

“hunger for affection” is colloquial style

163

“related to” -> are related to / relate to

165

Spelling varies: Big-5, big five

167ff.

2.2 Inclusion/ exclusion criteria

Missing: search time span, study designs included (controlled, RCTs, observational, review, online survey, …), study design quality, minimum sample size, ...

191

Exclusion criterion: “not published in English (language of study’s authors)” -> could be mistaken for: excluded because study authors native language was not English

A § like this is missing and required:

2.3 Assessments

Did study participants wear a clinical diagnosis given by clinical experts? Were only standardized validated questionnaires and/or screening instruments employed for self-reports? Criteria for correlations: taken as reported in text, significance, minimum magnitude, own computations if no correlation was given in the study (Cohen’s d or f/ eta-squared in ANOVAs into r) or study excluded, …

Authors denote correlational relationships with words like “links”, “fits perfectly with” (line 297), which seems colloquial.

264

“the 4-facets of temperament” -> the 4 facets of temperament

270

“Yücel” would be the correct spelling

278 + 292

“reliable association” -> what do authors mean by “reliable”?

280

Table headings: Table 3 is actually Table 2, and Table 2 is actually Table 3. Unexplained abbreviations occur: ATQ, TCI.

292

“…revealed one reliable temperament, heritable, cluster (88% consistency; see Table 2)” -> 88% consistency criterion is nowhere explained

339

“stronger in effect size” - >  what do authors mean by “stronger”?

419

“These findings broadly support may allow to better understand…” -> These findings add to the body of data that may help to better understand…

420

“…even when the symptoms of substance abuse or CSBD are subclinical.” -> This would be another chapter and populations with diagnosed disorders should be studied separately from those with subclinical disorders. For many in the field, “subclinical” is just another word for an “auxiliary construction”.

Author Response

Dear Reviewer,

Thank you for your comments and Review which helped us significantly improve our manuscript entitled “Addictive personality"? A systematic review of the overlap between compulsive sexual behavior and substance abuse in personality”. In this revised version of the paper, we have incorporated all of the comments and suggestions made by Reviewers and we hope that the manuscript is suitable for the publication.

Reviewer-1:

Comment 1: The ACS referencing style is supported by the journal, but several spelling of journal names should be corrected (e.g. “Addictive behavior” ->  “Addictive Behavior”). DOI-digits would be embraced.

Response: We carefully edited spelling mistakes and updated the ACS referencing style.

Comment 2: “Addictive personality” is not explained. Further, nowhere in the text any “personality profile” is given. I am afraid this concept is only used in its colloquial sense. The heading suggests that the appropriateness of the concept would be examined, instead correlational coherencies are reported – without referencing any authority.

Response: We now change the title to "Common features in compulsive sexual behavior, substance use disorders, personality, temperament and attachment – a narrative review".

Comment 3: “Substance abuse” is not adequately used. Authors obviously understand this as an overarching term, but this would be the term “addiction” or rather “substance use disorder (SUD”). In DSM-5 it is “mild” or “moderate” or “severe” SUD, in DSM-IV it was “substance abuse” and “substance dependence”. In ICD-10 it was “harmful use” (pretty much the same as “abuse”) and “dependence syndrome” were sub-categories. So “substance abuse/ harmful use” would logically exclude “dependence”.

The literature search terms (line 183f.), again, indicate that search results are not affected by this; still the term “abuse” as used here might perplex readers (especially psychiatrists).

Response: We now refrain from using the term “substance use disorder (SUD”) as requested.

Comment 5: Review design and presentations do not match the standards of an international journal. Authors’ review is not a ‘systematic review’ in terms of the PRISMA statement and other articles in the journal (e.g. Giulia Orilisi, Marco Mascitti, Lucrezia Togni, et al. Oral Manifestations of COVID-19 in Hospitalized Patients. Int. J. Environ. Res. Public Health 2021, 18(23), 12511; doi.org/10.3390/ijerph182312511.)

For a systematic review, too many information is missing – especially inclusion criteria, study quality assessment (best with results table), and table of descriptive characteristics of included studies. (See details below.)

Response: We thank the reviewer for this reference and now include in the tables: "Design", " Sample (n, Sex)", "Mean Age (Year)" and "Measure".

Comment 6: Authors’ conclusions are not adequately supported by their data. This affects interpretation (internal validity): It should be evidenced that the found correlations are specific for patients with compulsive sexual behavior (CSBD) and patients with substance use disorders (SUD) in contrast with non-clinical populations. And does the substance consumed make no difference? The specific substance/ substance group may be a moderator (sedatives, stimulants, multiple substance use).

Response: CSBD is a new field of research, and as such, there is only one paper that calculated correlations between big five on patients with compulsive sexual behavior (CSBD)- (Zilberman et al., 2018). We think there is still not  enough information for a true comparison to a non-clinical population  However, ourstudy attempts to establish compulsive sexual behavior as part of the addiction disorder. As mentioned, the debate on the issue is still ongoing.

Comment 7: Generalizability (external validity) of their results: “Do all addictions share common traits of an ‘addictive personality’ or do different addictions have distinct personality profiles?” – So starts the Abstract, but that is not what the text is about. Logical failure: Authors only study substance-related addiction and leave out all behavior-related ones. Thus, they can never refer to “all” addictions. The proposition in the Discussion section (line 355f.) is therefore over-interpreted: “…these personality-related dispositions seem to be at a core of a ‘addictive personality’.”

Response: We omitted the word "all" in the abstract and change the sentence to "Personality-related tendencies seem to shed further light on the meaning of "addictive personality".

Comment 8: After all of this, the heading is wrong: “Common features in compulsive sexual behavior, substance use disorders, personality, temperament and attachment – a narrative review” would have been more appropriate.

 Response: We now change the title as request.

Specific Comments

Spelling of the heading is incorrect (uses no capitals).

Response: Changed as requested.

Affiliations of authors are only vaguely given: Which faculty or department? What is this Interdisciplinary Center? E-Mail address of authors usually is their institutional address.

Response: Changed as requested.

After table notes should be an empty line.

Response: Changed as requested.

Parts 245-262, 283-302, and § ‘3.3 Commonalities and differences in attachment orientations’ do not report results. They belong to the Discussion section.

Response: Now we report and add more sentence in the Discussion section.

Parts ‘4.1 Future Directions’ and ‘4.2 Therapeutic Implications of the Current Research’ are interesting per se but are beyond the scope of the article.

Response: We omitted these parts as request.

Authors should limit the included studies to controlled studies to enhance internal validity of their findings (see Broad Comments f). Then they will have not 70 (but rather 17) studies to include.

Response: We understand your concern. The number of present studies is intentionally large because we want to examine three variables: big five, temperament and attachment.

The magnitudes of correlations are of SOME interest and should be reported in an international academic journal. 

Response: Now we report and add correlations.  

Line

Text / remarks / suggestions / proposals

39

“forms addictions” -> forms of addictions

Response: Changed as requested.

67

“classify  as” -> classified  as

Response: Changed as requested.

74f.

Reference?

Response: Now we add as requested.

86

“natural language” -> spoken language?

Response: Research use this term by theoretical of big five. For example: " One starting place for a shared taxonomy is the natural language of personality description. Beginning with Klages (1926), Baumgarten (1933), and Allport and Odbert (1936), various psychologists have turned to the natural language as a source of attributes for a scientific taxonomy".

100-130

Names of constructs/ traits/ attributes are sometimes written in capitals,

sometimes not. In the rest of the text no capital letters were used for this.

Response: Now we edit as requested.

148

“hunger for affection” is colloquial style

Response: Now we change the term to "Need".

163

“related to” -> are related to / relate to

Response: Now we change as requested

165

Spelling varies: Big-5, big five

Response: Now we change as requested

167ff.

2.2 Inclusion/ exclusion criteria

Missing: search time span, study designs included (controlled, RCTs, observational, review, online survey, …), study design quality, minimum sample size, ...

Response: Now we add as requested.

191

Exclusion criterion: “not published in English (language of study’s authors)” -> could be mistaken for: excluded because study authors native language was not English

Response: Changed, appears under inclusion criteria as “published in the English language”.

A § like this is missing and required:

2.3 Assessments

Did study participants wear a clinical diagnosis given by clinical experts? Were only standardized validated questionnaires and/or screening instruments employed for self-reports? Criteria for correlations: taken as reported in text, significance, minimum magnitude, own computations if no correlation was given in the study (Cohen’s d or f/ eta-squared in ANOVAs into r) or study excluded, …

Response: We now added a column to the tables that report on 'Assessments'.

Authors denote correlational relationships with words like “links”, “fits perfectly with” (line 297), which seems colloquial.

Response: Now we change as requested-

264

“the 4-facets of temperament” -> the 4 facets of temperament

Response: Now we change as requested

270

“Yücel” would be the correct spelling

Response: Now we change as requested

278 + 292

“reliable association” -> what do authors mean by “reliable”?

Response: We omitted the word “reliable”

280

Table headings: Table 3 is actually Table 2, and Table 2 is actually Table 3. Unexplained abbreviations occur: ATQ, TCI.

Response: We apologize for that and now the table in the right place. Moreover, we add the meaning of ATQ, TCI.

292

“…revealed one reliable temperament, heritable, cluster (88% consistency; see Table 2)” -> 88% consistency criterion is nowhere explained

Response: We omitted these parts as request.

339

“stronger in effect size” - >  what do authors mean by “stronger”?

Response: Now we change as requested.

419

“These findings broadly support may allow to better understand…” -> These findings add to the body of data that may help to better understand…

Response: Now we change as requested

420

“…even when the symptoms of substance abuse or CSBD are subclinical.” -> This would be another chapter and populations with diagnosed disorders should be studied separately from those with subclinical disorders. For many in the field, “subclinical” is just another word for an “auxiliary construction”.

Response: Now we change as requested- "below the clinical threshold"

Reviewer 2 Report

In this manuscript, the authors conducted a systematic review of the common and distinct personality traits between substance abuse and compulsive sexual behavior. The topic is interesting and the introduction well-written. The methods and results, however, need to be improved in order to generate reliable conclusions from the systematic review.

1), regarding the inclusion and exclusion criteria, the authors did not really define the inclusion criteria. Furthermore, what about subjects with comorbid psychiatric disorders?

2), for the results of the reviewed studies, not only significance, effect size measures such as correlation coefficient should be reported together. Other important information such as demographic information and sample size should also be reported for each study.

3), for the synthesis of the findings, do sex, age, and other important factors moderate the association between substance abuse/CSBD and personality? if there are data available, this should be mentioned and discussed.

4), some expressions should be revised or improved. For instance, line 14 in the abstract, perhaps add a "both" before people with a substance abuse; line 81, and/or should be "and"; line 167, "or" should be changed to "and those with" for instance.

Author Response

Dear Reviewer,

Thank you for your comments and Review which helped us significantly improve our manuscript entitled “Addictive personality"? A systematic review of the overlap between compulsive sexual behavior and substance abuse in personality”. In this revised version of the paper, we have incorporated all of the comments and suggestions made by Reviewers and we hope that the manuscript is suitable for the publication.

Reviewer-2:

Comment 1: ,regarding the inclusion and exclusion criteria, the authors did not really define the inclusion criteria. Furthermore, what about subjects with comorbid psychiatric disorders?

Response: Now, we have defined the inclusion criteria. We did not screen subjects with comorbid psychiatric disorders- In addition, we have added a section on the exclusion criteria.

Comment 2: ,for the results of the reviewed studies, not only significance, effect size measures such as correlation coefficient should be reported together. Other important information such as demographic information and sample size should also be reported for each study.

Response: We now report and added in the tables- sample size and gender, age and effect size as requested.

Comment 3: , for the synthesis of the findings, do sex, age, and other important factors moderate the association between substance abuse/CSBD and personality? if there are data available, this should be mentioned and discussed.

Response: We appreciate you raising this point. Unfortunately, we believe it’s beyond the scope of the current paper, but have noted in the future directions section of the paper.

Comment 4: some expressions should be revised or improved. For instance, line 14 in the abstract, perhaps add a "both" before people with a substance abuse; line 81, and/or should be "and"; line 167, "or" should be changed to "and those with" for instance.

Response: Changed as requested.

Round 2

Reviewer 1 Report

Manuscript ID: ijerph-1480489 / Review round 2

Common features in compulsive sexual behavior, substance use disorders, personality, temperament and attachment – a narrative review

Efrati, Kraus, Kaplan

(Modified heading; formerly "Addictive personality"? A systematic review of the overlap between compulsive sexual behavior and substance abuse in personality)

Broad Comments

a)     I am happy that authors understood all of my comments as constructive efforts to improve the manuscript. Now it is nearing a form that makes it ready for being published in this journal.

b)     I thank for the information c/o “natural language”. I learnt that this a technical linguistic term.

c)     It is good that authors do not claim anymore to study something like “the addictive personality” (one exception see line 378 b).

d)     The heading is now well chosen. Terminology is much more concise. Standards for a review are “OK” met. In their overworked form, tables now add substantially to an understanding.

e)     Authors obviously are unwilling to report amounts of correlation coefficients. They should be aware that this will downgrade the review in the eyes of the seasoned clinician and researcher. The information is just not the same:

Scale                      A          B          C         

Coefficients             .25       .50       .00

Symbols                 +          +          ─

f)      There are still many “links” etc. where the term “association, correlation, …” would be more appropriate. Again, authors should be aware that a colloquial style like this could downgrade the review in the eyes of the seasoned clinician and researcher  

Specific Comments

Authors wrote in a former response to this reviewer: “We think there is still not enough information for a true comparison to a non-clinical population”. This is an important result, they revealed desiderata for further research:

·       The article is not gender-sensitive and cannot be ─ because of the designs of included studies, as one can tell now from the information given in the tables. Correlation patterns will surely differ in m/f/d. In the Discussion section, a remark could be made that future research on this topic should reconsider gender aspects.

·       The included primary studies do not contrast clinical vs. non-clinical samples, like authors noted themselves. Correlation patterns could differ in clinical and non-clinical samples as well. Again, a remark could be made that future studies may focus on possible differences in different populations.

·       Another neglected research topic may be differences in correlational patterns between CSBD, personality dispositions, attachment style, and temperament depending on the substance group (users of stimulants, sedatives, alcohol, …).

The spelling of “big five personality” / “Big five personality” /… still varies… This is what my word processor tells me.

Line

Text / remarks / suggestions / proposals

Table 2

TCI proper name please with capital spelling: Temperament and Character Inventory (like in Table 3)

“Addicted group” is no measure, my suggestion: “expert assessment (not further specified)”

S.D. -> SD (like everywhere else in the text)

Methenamene -> Metamphetamine

PRISM -> please give full name, too

Table 3

I-CSB -> please give full name, too

202

“Drug” and “sex” are no groups. “Both groups score low on agreeableness [drug- 36; sex- 38] and conscientiousness [drug- 40; sex- 42]…” Do authors mean:

“Both groups score low on agreeableness [SUD ≤ 36; CSBD ≤ 38] and conscientiousness [SUD ≤ 40; CSBD ≤ 42] and high on neuroticism [SUD ≥ 63; CSBD ≥ 67]…”

378       

These personality-related dispositions seem to shed further light on the meaning of "addictive personality".

Here is the term again that has been fortunately avoided in the revised text... The sentence makes no good sense in a study which does not aim at any "addictive personality" any more. Do authors mean something like:
These correlational clusters may shed further light on our understanding of the psychology of addictive behaviors.

Author Response

Dear Reviewer,

Thank you for your comments and Review which helped us significantly improve our manuscript entitled “Common features in compulsive sexual behavior, substance use disorders, personality, temperament, and attachment – a narrative review”. In this revised version of the paper (round 2), we have incorporated all the comments and suggestions made by Reviewers and we hope that the manuscript is suitable for the publication.

Reviewer-1:

Comment 1: I am happy that authors understood all of my comments as constructive efforts to improve the manuscript. Now it is nearing a form that makes it ready for being published in this journal.

Response: We would like to thank the reviewer for his comments and for his kind words.

Comment 2: I thank for the information c/o “natural language”. I learnt that this a technical linguistic term.

Response: Thank you

Comment 3: It is good that authors do not claim anymore to study something like “the addictive personality” (one exception see line 378 b).

Response: We now change as request.

Comment 4: The heading is now well chosen. Terminology is much more concise. Standards for a review are “OK” met. In their overworked form, tables now add substantially to an understanding.

Response: We would like to thank the reviewer for his comments

Comment 5: Authors obviously are unwilling to report amounts of correlation coefficients. They should be aware that this will downgrade the review in the eyes of the seasoned clinician and researcher. The information is just not the same:

Scale                      A          B          C         

Coefficients             .25       .50       .00

Symbols                 +          +          ─

Response: Now we have reported the correlation coefficients in the Tables 1-3.

Comment 6: There are still many “links” etc. where the term “association, correlation, …” would be more appropriate. Again, authors should be aware that a colloquial style like this could downgrade the review in the eyes of the seasoned clinician and researcher  

Response: We carefully edited and changed as request

Specific Comments

Comment 7: Authors wrote in a former response to this reviewer: “We think there is still not enough information for a true comparison to a non-clinical population”. This is an important result, they revealed desiderata for further research:

Response: We agree. Thank you.

Comment 8: The article is not gender-sensitive and cannot be ─ because of the designs of included studies, as one can tell now from the information given in the tables. Correlation patterns will surely differ in m/f/d. In the Discussion section, a remark could be made that future research on this topic should reconsider gender aspects.

Response: We agree. We have added this point to the discussion. See lines 388-390.

Comment 9: The included primary studies do not contrast clinical vs. non-clinical samples, like authors noted themselves. Correlation patterns could differ in clinical and non-clinical samples as well. Again, a remark could be made that future studies may focus on possible differences in different populations.

Response: We agree. We have added this point on line 380-381.

Comment 10: Another neglected research topic may be differences in correlational patterns between CSBD, personality dispositions, attachment style, and temperament depending on the substance group (users of stimulants, sedatives, alcohol, …).

Response: We agree. We have added new language on lines 395-399.

Comment 11: The spelling of “big five personality” / “Big five personality” /… still varies… This is what my word processor tells me.

Response: We carefully edited and changed as request. We use 'Big five' only in the "Keywords" section.

Line

Text / remarks / suggestions / proposals

Comment 11: TCI proper name please with capital spelling: Temperament and Character Inventory (like in Table 3)

Response: Changed as request

Comment 12: “Addicted group” is no measure, my suggestion: “expert assessment (not further specified)”

Response: Changed as request

Comment 12: S.D. -> SD (like everywhere else in the text)

Response: Changed as request

Comment 13: Methenamene -> Metamphetamine

Response: Changed as request

Comment 14: PRISM -> please give full name, too

Response: Changed as request

Table 3

Comment 15: I-CSB -> please give full name, too

Response: Changed as request

202

Comment 16: “Drug” and “sex” are no groups. “Both groups score low on agreeableness [drug- 36; sex- 38] and conscientiousness [drug- 40; sex- 42]…” Do authors mean:

“Both groups score low on agreeableness [SUD ≤ 36; CSBD ≤ 38] and conscientiousness [SUD ≤ 40; CSBD ≤ 42] and high on neuroticism [SUD ≥ 63; CSBD ≥ 67]…”

Response: We add to the Tables

378       

Comment 17: These personality-related dispositions seem to shed further light on the meaning of "addictive personality".

Here is the term again that has been fortunately avoided in the revised text... The sentence makes no good sense in a study which does not aim at any "addictive personality" any more. Do authors mean something like:
These correlational clusters may shed further light on our understanding of the psychology of addictive behaviors.

Response: Changed as request

Reviewer 2 Report

Thank the authors for addressing my concerns.

1), what is the rationale for using the inclusion criterion of "including 27 subjects or more for a clinical sample and 150 or more for a non-clinical sample"? why 27 and 150?

2), for the inclusion criterion "based on a quantitative research design", please specify what kind of quantitative design.

3), for Tables 1-3, the authors should report the actual coefficient for the correlations, and for comparison studies, if cohen's d for instance is available, they should also report. That is what I meant by reporting effect size measures. Right now, they only denoted - or +. I do appreciate that the authors added the description of subject characteristics.

Author Response

Dear Reviewer,

Thank you for your comments and Review which helped us significantly improve our manuscript entitled “Common features in compulsive sexual behavior, substance use disorders, personality, temperament, and attachment – a narrative review”. In this revised version of the paper (round 2), we have incorporated all of the comments and suggestions made by Reviewers and we hope that the manuscript is suitable for the publication.

Reviewer-2:

Comment 1: what is the rationale for using the inclusion criterion of "including 27 subjects or more for a clinical sample and 150 or more for a non-clinical sample"? why 27 and 150?

Response: Studies in the clinical population are based on treatment groups of addicts. The treatment groups are between 25 and 30 patients, so we chose the number 27. In the non-clinical population, we chose a much higher number out of the need to maintain an appropriate statistical level (n=150).

Comment 2: for the inclusion criterion "based on a quantitative research design", please specify what kind of quantitative design.

Response: In the Tables we clarified the specify kind of quantitative design.

Comment 3: for Tables 1-3, the authors should report the actual coefficient for the correlations, and for comparison studies, if cohen's d for instance is available, they should also report. That is what I meant by reporting effect size measures. Right now, they only denoted - or +. I do appreciate that the authors added the description of subject characteristics.

Response: Now we report and add correlations to 1-3 Tables as request.